# A Persistent Spatial Semantic Representation for High-level Natural Language Instruction Execution

**Valts Blukis**[1,2], **Chris Paxton**[1], **Dieter Fox**[1,3], **Animesh Garg**[1,4], **Yoav Artzi**[2]
[1]NVIDIA        [2]Cornell University        [3]University of Washington
[4]University of Toronto, Vector Institute

**Abstract:** Natural language provides an accessible and expressive interface to specify long-term tasks for robotic agents. However, non-experts are likely to specify such tasks with high-level instructions, which abstract over specific robot actions through several layers of abstraction. We propose that key to bridging this gap between language and robot actions over long execution horizons are persistent representations. We propose a persistent spatial semantic representation method, and show how it enables building an agent that performs hierarchical reasoning to effectively execute long-term tasks. We evaluate our approach on the ALFRED benchmark and achieve state-of-the-art results, despite completely avoiding the commonly used step-by-step instructions. https://hlsm-alfred.github.io/

**Keywords:** vision and language, spatial representations

## 1 Introduction

Mobile manipulation in a home environment requires addressing multiple challenges, including exploration and making long-term inference about actions to perform. In addition to reasoning, robots require an accessible, yet sufficiently expressive interface to specify their tasks. Natural Language provides an intuitive mechanism for task specification, and coupled with advances in automated language understanding, is increasingly applied to embodied agents [e.g., 1–11].

In this paper, we study the problem of learning to map high-level natural language instructions to low-level mobile manipulation actions in an interactive 3D environment [12]. Existing work largely studies language tightly aligned to the robot actions, either using single-sentence instructions [e.g., 1, 2, 5, 9] or sequences of instructions [13–18]. In contrast, we focus on high-level instructions, which provide more efficient human-robot communication, but require long-horizon reasoning across layers of abstraction to generate actions not explicitly specified in the instruction.

Robust reasoning about manipulation goals from unrestricted high-level natural language instructions has a variety of open challenges. Consider the instruction *secure two discs in a bedroom safe* (Figure 1). The robot must first locate the *safe* in the *bedroom*. It then needs to distribute the actions entailed by *secure* to two objects (*two discs*), each requiring a distinct sequence of actions, but targeting the same *safe*. It is also required to map the verb *secure* to its action space. In parallel, the robot must address mobile manipulation challenges, and often can only identify required actions as it observes and manipulates the world (e.g., if the *safe* needs to be opened).

We propose to construct and continually update a spatial semantic representation of the world from robot observations (Figure 2). Similar to widely used map representations [19–22], we retain the spatial properties of the environment, allowing the robot to navigate and reason about relations between objects, as required to accomplish its task. We propose the Hierarchical Language-conditioned Spatial Model (HLSM), a hierarchical approach that uses our spatial representation as a long-term memory to solve long-horizon tasks. HLSM consists of a high-level controller that generates sub-goals, and a low-level controller that generates sequences of actions to accomplish them. In our example (Figure 1), the sequence of subgoals is ⟨pick up a CD, open the safe, put the CD in the safe, . . . ⟩, each requiring a sequence of actions. The spatial representation allows selecting subgoals that use previously observed objects outside of the agent's view, or to decide about needed exploration.

We evaluate our approach on the ALFRED [12] benchmark and achieve state-of-the-art results without using the low-level instructions used by previous work [16–18, 23], neither during training

5th Conference on Robot Learning (CoRL 2021), London, UK.

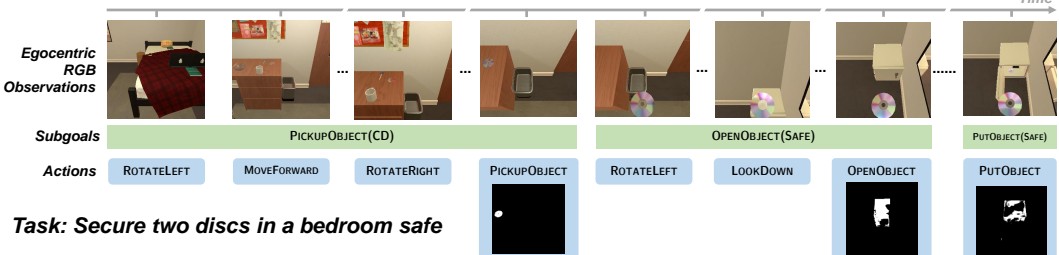

**Figure 1:** Illustration of the task and our hierarchical formulation. The agent receives a high-level task in natural language. It needs to map RGB images to navigation and manipulation actions to complete the task.

nor at test-time. This paper makes three key contributions: (a) a modular representation learning approach for the problem of mapping high-level natural language task descriptions to actions in a 3D environment; (b) a method for utilizing a spatial semantic representation within a hierarchical model for solving mobile manipulation tasks; and (c) state-of-the-art performance on the ALFRED benchmark, even outperforming all approaches that use detailed sequential instructions.

## 2 Related Work

Natural language has been extensively studied in robotics research, including with focus on instruction [1, 24], reference resolution [25], question generation [26–28], and dialogue [4, 29, 30]. Most work in this area has considered either synthetic instructions of relatively simple goals [7, 31–33], or natural language instructions where all intermediate steps are explained in detail [5, 12–14, 34–38]. In contrast, we focus on high-level instructions, which are more likely in home environments [39].

Representation of world state, action history, and language semantics plays a central role in robot systems and their algorithm design. Symbolic representations have been extensively studied for instruction following agents [1–4, 19, 20, 39–45]. While they simplify the symbol grounding problem and enable robustness, the ontologies on which they rely on are laborious to scale to new, unstructured environments and language. Representation learning presents an alternative by learning to map observations and language directly to actions [5, 8, 9, 11, 13, 34]. World state and language semantics are represented with vectors [13] or by memorizing past observations [8, 17]. Modelling improvements have enabled these approaches to achieve good performance on complex navigation tasks [7, 9, 11, 13, 14, 37], a success that has not yet translated to mobile manipulation [12, 46, 47].

We propose integrating a semantic voxel map state representation within a hierarchical representation learning system. Similar semantic 2D maps have been successfully used in navigation [7, 8, 48, 49] and more recently even in mobile manipulation instruction-following tasks [23]. We extend these maps to 3D and show state-of-the-art results on a challenging mobile manipulation benchmark. Our map design is related to sparse metric, topological and semantic maps [10, 19–21, 50] that have enabled grounding symbolic instruction representations. Our map does not impose a topological structure or require reasoning about object instances, instead modelling a distribution over semantic classes for every voxel.

## 3 Problem Definition

Let $\mathcal{A}$ be the set of agent actions, and $\mathcal{S}$ the set of world states. Given a natural language instruction $L$ and an initial state $s_0 \in \mathcal{S}$, the agent's goal is to generate an execution $\Xi = \langle s_0, a_0, s_1, a_1, \ldots, s_T, a_T \rangle$, where $a_t \in \mathcal{A}$ is an action taken by the agent at time $t$, $s_t \in \mathcal{S}$ is the state before taking $a_t$, and $s_{t+1} = \mathcal{T}(s_t, a_t)$ under environment dynamics $\mathcal{T} : \mathcal{S} \times \mathcal{A} \to \mathcal{S}$. The state $s_t$ is defined by the environment layout and the poses and states of all objects and the agent. The agent does not have access to the state $s_t$, but only to an observation $o_t$. An observation $o_t = (I_t, P_t, v_t^S, L)$ includes a first-person RGB camera image $I_t$, the agent's pose $P_t$, a one-hot encoding of the object class the agent is holding $v_t^S$, and the instruction $L$. The task is considered successful if all goal-conditions corresponding to the task $L$ are true at the final state $s_T$. Partial success is measured as the fraction of goal-conditions that have been achieved.

The ALFRED dataset includes sets of seen and unseen environments. The set of actions $\mathcal{A} = \mathcal{A}_{\mathrm{nav}} \cup \mathcal{A}_{\mathrm{int}}$ includes parameter-free navigation actions $\mathcal{A}_{\mathrm{nav}} = \{\text{MOVEAHEAD}, \text{ROTATELEFT}, \text{ROTATERIGHT}\}$ and interaction actions $\mathcal{A}_{\mathrm{int}} =$

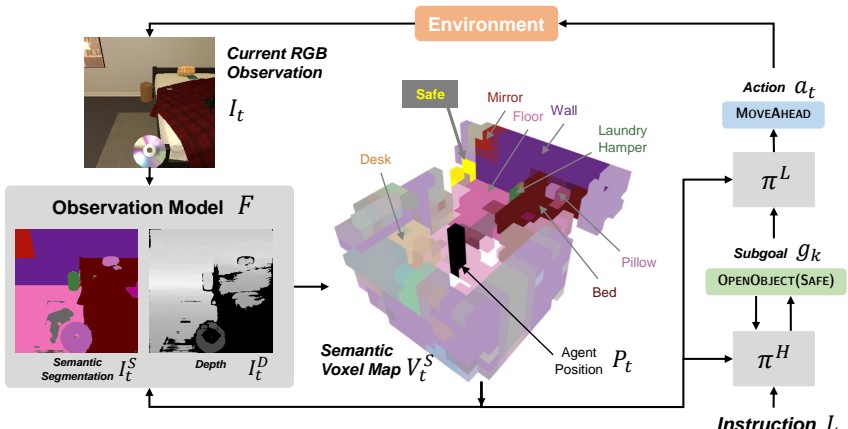

**Figure 2:** Model architecture consisting of an observation model, high-level controller ($\pi^H$), and low-level controller ($\pi^L$). The observation model updates the semantic voxel map state representation from RGB observations. $\pi^H$ predicts the next subgoal given the instruction and the map. $\pi^L$ outputs a sequence of actions to achieve the subgoal. The semantic voxel map is visualized in the middle with agent position illustrated as a black pillar, ans the current sugoal argument mask in yellow. Other colors are different segmentation classes. Saturated voxels are observed in the current timestep.

{PICKUP, PUT, TOGGLEON, TOGGLEOFF, OPEN, CLOSE, SLICE} parameterized by a binary mask that identifies the object of the interaction in the agent's current first-person view. We compute $P_t$ and $v_t^S$ using dead-reckoning from RGB observations and actions.

## 4 Hierarchical Model with a Persistent Spatial Semantic Representation

We model the agent behavior with a policy $\pi$ that maps an instruction $L$ and the observation $o_t$ at time $t$ to an action $a_t$. The policy $\pi$ is made of an *observation model* $F$ and two controllers: a *high-level controller* $\pi^H$ and a *low-level controller* $\pi^L$. The observation model builds a spatial *state representation* $\hat{s}_t$ that captures the cumulative agent knowledge of the world at time $t$. $\hat{s}_t$ is used by both $\pi^H$ for high-level long-horizon task planning, and $\pi^L$ for near-term reasoning, such as object search, navigation, collision avoidance, and manipulation. Figure 2 illustrates the policy.

The high-level controller $\pi^H$ computes a probability over *subgoals*. A subgoal $g$ is a tuple $(\texttt{type}, \texttt{arg}^C, \texttt{arg}^M)$, where $\texttt{type} \in \mathcal{A}_{\text{int}}$ is an interaction type (e.g., OPEN, PICKUP), $\texttt{arg}^C$ is the semantic class of the interaction argument (e.g., SAFE, CD), and $\texttt{arg}^M$ is a 3D mask identifying the location of the argument instance. In ALFRED, each interaction action in the set $\mathcal{A}_{\text{int}}$ corresponds to a subgoal type. When predicting the $k$-th subgoal at time $t$, $\pi^H$ considers the instruction $L$, the current state representation $\hat{s}_t$, and the sequence of past subgoals $\langle g_i, \rangle_{i<k}$. During inference, we sample from $\pi^H$. Unlike $\arg\max$, sampling allows the agent to re-try the same or different subgoal incase of a potentially random failure (e.g., if a MUG was not found, pick up a CUP).

The low-level controller $\pi^L$ is given the subgoal $g_k$ as its goal specification at time $t$. At every timestep $j > t$, $\pi^L$ maps the state representation $\hat{s}_j$ and subgoal $g_k$ to an action $a_j$, until it outputs one of the stop actions: $a_{\text{PASS}}$ or $a_{\text{FAIL}}$ to indicate successful or failed subgoal completion.

The execution flow is as follows. At time $t = 0$ the initial observation $o_0$ is received. At each timestep, we update the state representation $\hat{s}_t$ using the observation model. If there is no currently active subgoal, we sample a new subgoal $g_k$ from $\pi^H$, and then sample an action $a_t$ from $\pi^L$. If $a_t$ is $a_{\text{PASS}}$, we increment subgoal counter $k$. If it is $a_{\text{FAIL}}$, we discard the current subgoal $k$. We repeat sampling subgoals and actions until an executable action $a_t$ is sampled. We execute $a_t$, increment the timestep $t$, and receive the next observation $o_t$. The episode ends when the subgoal $g_{\text{STOP}}$ is sampled or the horizon $T_{max}$ is exceeded. Algorithm **??** in Appendix **??** describes this process.

### 4.1 State Representation

The state representation $\hat{s}_t$ at time $t$ captures the agent's current understanding of the state of the world, including the locations of objects observed and the agent's relation to them. The state representation is a tuple $(V_t^S, V_t^O, v_t^S, P_t)$. The semantic map $V_t^S \in [0, 1]^{X \times Y \times Z \times C}$ is a 3D voxel map that for every position indicates which of the $c \in [1, C]$ object classes are present in the voxel. The

observability map $V_t^O \in \{0,1\}^{X \times Y \times Z}$ is a 3D voxel map that indicates whether the corresponding position has been observed. The inventory vector $v_t^S \in \{0,1\}^C$ indicates which of the $C$ object classes the agent is currently holding. The agent pose $P_t = (x, y, \omega_p, \omega_y)$ is specified by the 2D position $(x, y)$, pitch angle $\omega_p$, and yaw angle $\omega_y$.

We also compute 2D *state affordance features* $\text{AFFORD}(\hat{s}_t) \in [0,1]^{7 \times X \times Y}$ in a top-down view that represent each position with one or more of seven affordance classes {pickable, receptacle, togglable, openable, ground, obstacle, observed}. Each $[\text{AFFORD}(\hat{s}_t)]_{(\tau, x, y)} = 1.0$ if at least one of the voxels at position $(x, y)$ has affordance class $\tau$, otherwise it is zero. $\text{AFFORD}(\hat{s}_t)$ is suited for object class agnostic reasoning, for example predicting a pose to pick up an object. [1]

## 4.2 Observation Model

The observation model $F(\hat{s}_{t-1}, o_t, g_k)$ updates the state representation with new observations. It considers the current subgoal $g_k$ to actively acquire information relevant to $g_k$. The computation of $F$ consists of three steps: perception, projection, accumulation.

**Perception Step** We predict semantic segmentation $I_t^S$ and depth map $I_t^D$ from the RGB observation $I_t$. We use neural networks pre-trained in the ALFRED environment. The semantic segmentation $[I_t^S]_{(u,v)}$ is a distribution over $C$ object classes at pixel $(u, v)$. The depth map $[I_t^D]_{(u,v)}$ is a binned distribution over $B$ bins. [2] We also heuristically compute a binary mask $M_t^D$ that indicates which pixels have confident depth readings. We allow more confidence slack in pixels that correspond to the current subgoal argument $\text{arg}_t^C$ according to $I_t^S$. Appendix ?? provides further details. We use perception models based on the U-Net [51] architecture, but our framework supports other, potentially more powerful models as well (e.g. [52, 53]).

**Projection Step** We use a pinhole camera model to convert depth $I_t^D$ and segmentation $I_t^S$ to a point cloud that represents each image pixel $(u, v)$ with a 3D position $(x, y, z) \in \mathbb{R}^{X \times Y \times Z}$ and a semantic distribution $[I_t^S]_{(u,v)}$. We use $\arg\max_B(I_t^D)$ to compute the 3D positions, and discard points at pixels $(u, v)$ when the binary mask value is $[M_t^D]_{(u,v)} = 0$. We construct a discrete semantic voxel map $\hat{V}_t^S \in [0,1]^{X \times Y \times Z \times C}$, where $X$, $Y$, and $Z$ are the width, height, and length. The value at each voxel $[\hat{V}_t^S]_{(x,y,z)}$ is the element-wise maximum of the segmentation distributions $[I_t^S]_{(u,v)}$ across all points $(u, v)$ within the voxel. We additionally compute a binary observability map $\hat{V}_t^O \in \{0,1\}^{X \times Y \times Z}$ that indicates the voxels observed at time $t$. A voxel is observed if it contains points, or if a ray cast from the camera through the voxel centroid has expected depth greater than the distance from the camera to the centroid.

**Accumulation Step** We integrate $\hat{V}_t^S$ and $\hat{V}_t^O$ into a persistent state representation:

$$V_t^S = \hat{V}_t^S \times \hat{V}_t^O + V_{t-1}^S \times (1 - \hat{V}_t^O) \qquad V_t^O = \max(V_{t-1}^O, \hat{V}_t^O) \ . \qquad (1)$$

This operation updates each voxel with the most recent semantic distribution, while retaining the values of all voxels not visible at time $t$. The output of the observation model is the spatial state representation $\hat{s}_t = (V_t^S, V_t^O, v_t^S, P_t)$. The inventory $v_t^S$ and pose $P_t$ are taken directly from $o_t$.

## 4.3 High-level Controller ($\pi^H$)

At timestep $t$, when invoked for the $k$-th time, the input to $\pi^H$ is the instruction $L$, the sequence of past subgoals $\langle g_i \rangle_{i<k}$, and the current state representation $\hat{s}_t$. The output is the next subgoal $g_k = (\text{type}_k, \text{arg}_k^C, \text{arg}_k^M)$. Figure 3 illustrates the high-level controller architecture.

**Input Encoding** We encode the text $L$ using a pre-trained BERT [54] model that we fine-tune during training. We use the CLS token embedding as the task embedding $\phi^L$. We encode the state representation $\hat{s}_t$ to account for classes of all observed objects, and the object that the agent is holding: $\phi^s(\hat{s}_t) = [v_t^S; \max_{(x,y,z)}(V_t^S)]$, where $\max_{(x,y,z)}$ is a max-pooling operation over spatial dimensions and $[\cdot; \cdot]$ denotes concatenation. We compute the representations of previous subgoals as $\langle \text{REPR}(g_i) \rangle_{i=0}^{k-1}$, where $\text{REPR}(g_i)$ is the sum of a sinusoidal positional encoding [55] of index $i$ and

---

[1] We assume a known mapping between object semantic classes and affordance classes.

[2] We use $B$ uniformly spaced depth bins $\{0, \Delta_D, 2\Delta_D, \ldots, (B-1)\Delta_D\}$, where $\Delta_D$ is a depth resolution. We suggest $\Delta_D$ should be less than 50% of the voxel size. We used voxels with edge length 0.25m.

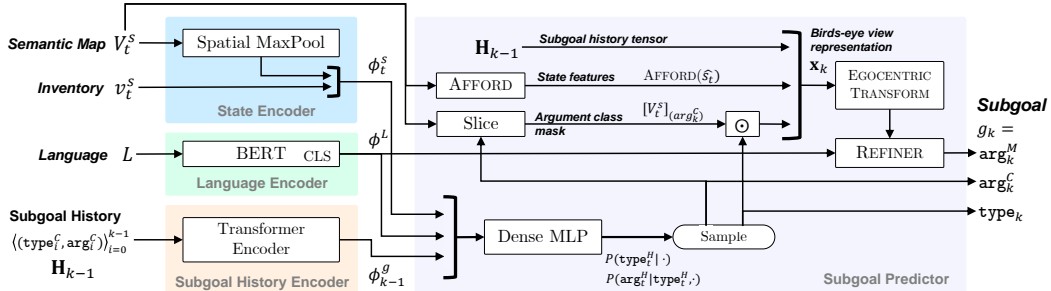

**Figure 3:** Illustration of the high-level controller $\pi^H$ (Section 4.3).

learned embeddings for $\texttt{type}_i$ and $\texttt{arg}_i^C$. We process this sequence with a two-layer Transformer autoregressive encoder [55] to compute $\langle\phi_i^g\rangle_{i=0}^{k-1}$. We take $\phi_{k-1}^g$ as the subgoal history embedding vector. We additionally encode the argument mask information $\texttt{arg}_i^M$ from the subgoal history in an integer-valued subgoal history tensor $\mathbf{H}_{k-1} \in \mathbb{N}^{K\times X\times Y}$ where $[\mathbf{H}_{k-1}]_{(\tau,x,y)}$ is the number of times an interaction action type $\tau$ was performed at 2D position $(x,y)$ in the birds-eye view:

$$[\mathbf{H}_{k-1}]_{(\tau,x,y)} = \sum_{\substack{i=0\ldots k-1 \\ \texttt{arg}_i^C=\tau}}^{k-1} \max_z([\texttt{arg}_i^M]_{(x,y,z)}) \ . \tag{2}$$

**Subgoal Prediction** We concatenate the three representations $\mathbf{h}_{(t,k)} = [\phi^L; \phi_t^s; \phi_{k-1}^g]$. We use a densely connected multi-layer perceptron [56] to predict two distributions $P(\texttt{type}_k \mid \mathbf{h}_{(t,k)})$ and $P(\texttt{arg}_k^C \mid \texttt{type}_k, \mathbf{h}_{(t,k)})$, from which we sample a subgoal type $\texttt{type}_k$ and argument class $\texttt{arg}_k^C$.

The remaining component of the subgoal is the action argument mask $\texttt{arg}_k^M$. Let $[V_t^S]_{(\texttt{arg}_k^C)}$ be a voxel map that only retains the object information for objects of class $\texttt{arg}_k^C$ in the semantic map $V_t^S$. We refine it to identify a single object instance. We compute a birds-eye view representation:

$$\mathbf{x}_t = [\textsc{Afford}(\hat{s}_t); \ \mathbf{H}_{k-1}; \ \max_z([V_t^S]_{(\texttt{arg}_k^C)}) \otimes \mathbb{1}_{\texttt{type}_k}] \tag{3}$$

where $\textsc{Afford}(\hat{s}_t)$ is a birds-eye view state affordance feature map (Section 4.1) and $\mathbb{1}_{\texttt{type}_k}$ is a one-hot encoding of $\texttt{type}_k$.[3] Finally, we compute the 3D argument mask $\texttt{arg}_k^M \in [0,1]^{X\times Y\times Z}$:

$$\texttt{arg}_k^M = \textsc{Refiner}(\textsc{EgoTransform}(\mathbf{x}_t, P_t), \phi^L) \ , \tag{4}$$

where $\textsc{EgoTransform}(\mathbf{x}, P_t)$ transforms the map $\mathbf{x}$ to the agent egocentric pose $P_t$, $\textsc{Refiner}$ is a neural network based on the LingUNet architecture [36], and $\phi^L$ is the language embedding. The refined $\texttt{arg}_k^M$ is a $[0,1]$-valued 3D mask that identifies the instance of the interaction argument object. If the object is believed to be unobserved, then $\texttt{arg}_k^M$ contains all zeroes. The controller output is the subgoal $g_k = (\texttt{type}_k, \texttt{arg}_k^C, \texttt{arg}_k^M)$.

## 4.4 Low-level Controller ($\pi^L$)

The low-level controller $\pi^L$ is conditioned on the most recent subgoal $g_k = (\texttt{type}_k, \texttt{arg}_k^C, \texttt{arg}_k^M)$. At time $t$, it maps the state representation $\hat{s}_t$ to an action $a_t$. It combines engineered and learned components. Appendix ?? provides the implementation details. The controller $\pi^L$ invokes a set of procedures: `NavigateTo`, `SampleExplorationPosition`, `SampleInteractionPose`, and `InteractMask`. Their invocation follows a pre-specified execution flow across multiple timesteps. First, we perform a 360° rotation to observe the nearby environment. If no objects of type $\texttt{arg}_k^C$ are observed, we explore the environment by sampling a position $(x,y) = $ `SampleExplorationPosition`$(\hat{s}_t)$, navigating there using the procedure `NavigateTo`$(x, y, \hat{s}_t)$, and performing a 360° rotation. We repeat exploration until a voxel in $V_t^S$ contains the class $\texttt{arg}_k^C$ with >50% probability. To interact with an object, we sample an interaction pose $(x, y, \omega_y, \omega_p) = $ `SampleInteractionPose`$(\hat{s}_t, g_k)$, invoke `NavigateTo`$(x, y, \hat{s}_t)$ to reach the position (x,y), and then rotate according to yaw and pitch angles $(\omega_y, \omega_p)$. Finally, we generate the egocentric interaction mask $\texttt{mask}_t = $ `InteractMask`$(\hat{s}_t, \texttt{arg}_k^M)$, and output the interaction action $(\texttt{type}_k, \texttt{mask}_t)$.

---

[3]$\otimes$ denotes multiplication of a $X \times Y$ tensor with a $K$-dimensional vector to obtain a $K \times X \times Y$ tensor. $[\cdot; \cdot; \cdot]$ denotes channel-wise concatenation.

All procedures use the spatial representation $\hat{s}_t$. NavigateTo navigates to a goal position using a value iteration network (VIN) [57] that reasons over obstacle and observability maps from $\hat{s}_t$. SampleExplorationPosition samples positions on the boundary of observed space in $\hat{s}_t$. SampleInteractionPose uses a learned neural network NavModel to predict a distributon of poses from which the interaction $g_k$ will likely succeed. InteractMask uses the segmentation image $I_t^S$ and the 3D argument mask $\arg_t^M$ to compute the first-person mask of the target object.

## 5 Learning

The policy contains four learned models: the segmentation and depth networks, $\pi^H$, and the navigation model NavModel used by $\pi^L$. We train all four networks independently using supervised learning. We assume access to a training dataset $\mathcal{D} = \{(L^{(j)}, \Xi^{(j)})\}_{j=1}^{N_D}$ of high-level natural language instructions $L^{(j)}$ paired with demonstration execution $\Xi^{(j)}$ in a set of seen environments. Each execution $\Xi^{(j)}$ is a sequence of states and actions $\langle s_0^{(j)}, a_0^{(j)}, \ldots, s_T^{(j)}, a_T^{(j)} \rangle$. We denote $N_P$ the total number of states in dataset $\mathcal{D}$, and $N_G$ the total number of subgoals.

We process $\mathcal{D}$ into three datasets. The perception dataset $\mathcal{D}^P = \{([I]^{(i)}, [I^D]^{(i)}, [I^S]^{(i)}\}_{i=1}^{N_P}$ includes RGB images $[I]^{(i)}$ with ground truth depth $[I^D]^{(i)}$ and segmentation $[I^S]^{(i)}$. The subgoal dataset $\mathcal{D}^g = \{(L^{(i)}, \hat{s}_t^{(i)}, \langle g_j^{(i)} \rangle_{j=0}^k)\}_{i=1}^{N_G}$ contains natural language instructions $L^{(i)}$, state representations $\hat{s}_t^{(i)}$ at the start of $k$-th subgoal execution, and sequences of the first $k$ subgoals $\langle g_j^{(i)} \rangle_{j=0}^k$ extracted from $\Xi^{(j)}$. The navigation dataset $\mathcal{D}^N = \{(\hat{s}^{(i)}, g^{(i)}, P^{(i)})\}_{i=1}^{N_P}$ consists of state representations $\hat{s}^{(i)}$, subgoals $g^{(i)}$, and agent poses $P^{(i)}$ at the time of taking the interaction action corresponding to subgoal $g^{(i)}$. The state representations $\hat{s}^{(\cdot)}$ in datasets $\mathcal{D}^g$ and $\mathcal{D}^N$ are constructed using the observation model (Section 4.2), but using ground-truth depth and segmentation images.

We train the perception models on $\mathcal{D}^P$ and the $\pi^H$ on $\mathcal{D}^g$ to predict the $k$-th subgoal by optimizing cross-entropy losses. We use $\mathcal{D}^N$ to train the navigation model NavModel by optimizing a cross-entropy loss for positions and yaw angles, and an L2 loss for the pitch angle.

## 6 Experimental Setup

**Environment, Data, and Evaluation** We evaluate our approach on the ALFRED [12] benchmark. It contains 108 training scenes, 88/4 validation seen/unseen scenes, and 107/8 test seen/unseen scenes. There are 21,023 training tasks, 820/821 validation seen/unseen tasks, and 1533/1529 test seen/unseen tasks. Each task is specified with a high-level natural language instruction. The goal of the agent is to map raw RGB observations to actions to complete the task. ALFRED also provides detailed low-level step-by-step instructions, which simplify the reasoning process. We do not use these instructions for training or evaluation. We collect a training dataset of language-demonstration pairs for learning (Section 5). To extract subgoal sequences, we label each interaction action $a_t = (\text{type}_t, \text{mask}_t)$ and any preceding navigation actions with a single subgoal of $\text{type} = \text{type}_t$. We compute the subgoal argument class $\arg^C$ and 3D mask $\arg^M$ labels from the first-person mask $\text{mask}_t$, and ground truth segmentation and depth. Completing a task requires satisfying several goal conditions. Following the common evaluation [58, 59], we report two metrics. *Success rate* (SR) is the fraction of tasks for which all goal conditions were satisfied. *Goal condition rate* (GC) is the fraction of goal-conditions satisfied across all tasks.

**Systems** We compare our approach, the Hierarchical Language-conditioned Spatial Model (HLSM) to others on the ALFRED leaderboard that only use the high-level instructions. At the time of writing, the only such published approach is HiTUT [47], an approach that uses a flat BERT [54] architecture to model a hierarchical task structure without using a spatial representation. See Appendix **??** for a detailed comparison. We also compare to approaches that use the step-by-step instructions, which puts our method at a disadvantage. Of these, LAV [60] also imposes a hierarchical task structure and uses pre-trained depth and segmentation models, but without using a spatial state representation.

Additionally, we perform ablations and study sensory oracles. To study the observation model, we compare to using sensory oracles for ground truth depth, ground truth segmentation, and both. We report high-level controller ablations that remove the subgoal encoder, language encoder, and state

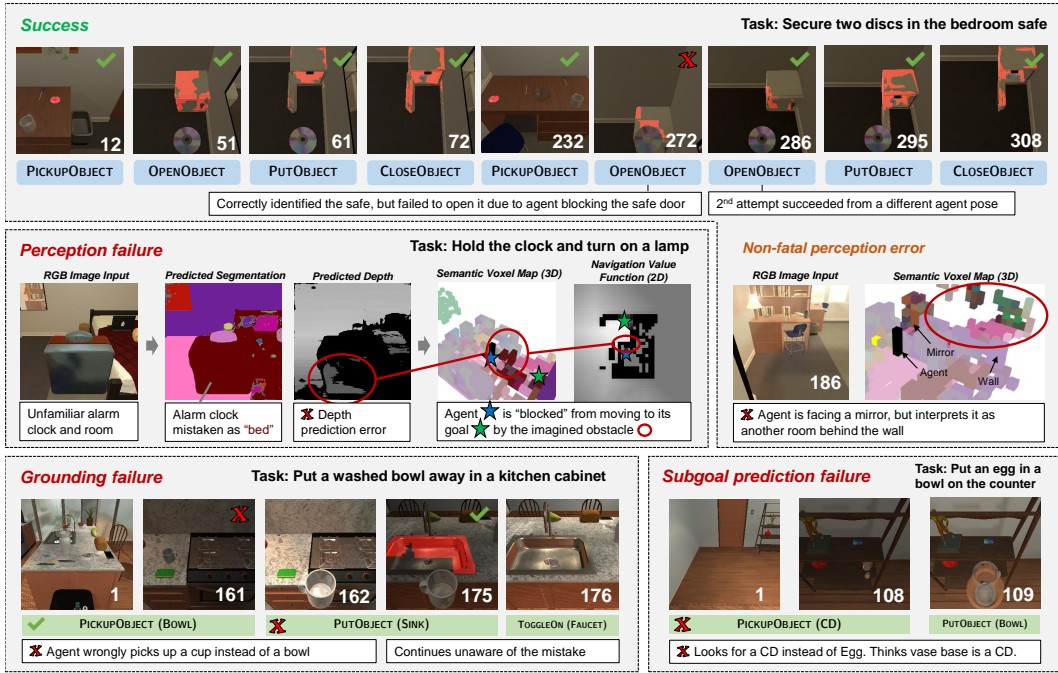

**Figure 4:** Qualitative results showcasing successes and failures of our approach. **Top row:** snapshots of every interaction action taken during a successful task. Action argument masks are overlaid in red over the RGB images. The white numbers are timesteps. **Middle-right:** illustration of a non-fatal perception error. **Middle-left:** illustration of a fatal perception error. The agent incorrectly interprets the reflection on the alarm clock as an obstacle, causing the agent (blue star) to believe that the path to the goal (green star) is blocked off. This is reflected in the navigation value function computed by the value iteration network (VIN) [57], where black cells are obstacles with value −1. White cell is the goal with value 1. **Bottom-left:** grounding failure. The agent wrongly picks up the cup instead of a bowl. Predicted subgoals are shown in green. **Bottom-right:** high-level controller and percepton failure. $\pi^H$ predicts the wrong subgoal argument class (CD instead of EGG). The segmentation model then mistakes the vase for a CD.

representation encoder as used for predicting subgoal type $\text{type}_k$ and argument class $\text{arg}_k^C$, while still using the state representation $\hat{s}_t$ to predict the subgoal argument mask $\text{arg}_k^M$. We also study a low-level controller ablation that removes the exploration procedure.

## 7 Results

Table 1 shows test and validation results. Our approach achieves state-of-the-art performance across both seen and unseen environments in the setting with only high-level instructions. We achieve 10.04% absolute (98.1% relative) improvement in SR on the test unseen split, and 11.53% absolute (62.6% relative) improvement in SR on the test seen split compared to HiTUT G-only.

Our approach performs competitively even when compared to approaches that also use the low-level step-by-step instructions. We achieve 4.84% absolute (31.4% relative) improvement in SR on the test unseen split compared to ABP [61]. On the test seen split, our approach performs reasonably well, however ABP [61] and LWIT [18] perform better, reflecting potentially stronger scene overfitting.

Tables 2 and 3 show development results. We performed five runs of the full HLSM model on the validation unseen data and found the sample standard deviation of the success rate is 1.1% (absolute). All other results are from a single-evaluation runs. Ground truth depth alone (+ gt depth) does not significantly affect performance. Ground truth segmentation (+ gt seg) provides 6.6%/16.4% absolute improvement in seen/unseen scenes. Using both (+ gt depth, gt seg) provides 11.1%/21.9% absolute improvement and narrows the seen/unseen gap from 11.3% to 0.5%. This points to perception being the main bottleneck in generalization to unseen scenes.

We report high-level controller $\pi^H$ input encoder ablations. The poor performance without the language encoder reflects task difficulty. Zeroing the input to the subgoal history encoder (but keeping position encodings) does not significantly affect performance, showing that knowing the index of the current subgoal in addition to the state representation is often sufficient. Not using the state representation for predicting subgoal type and argument class gives mixed results in seen

**Table 1**

| Method | Test | | | | Validation | | | |
|---|---|---|---|---|---|---|---|---|
| | Seen | | Unseen | | Seen | | Unseen | |
| | SR | GC | SR | GC | SR | GC | SR | GC |
| **Low-level Sequential Instructions + High-level Goal Instruction** | | | | | | | | |
| SEQ2SEQ [12] | 3.98 | 9.42 | 0.39 | 7.03 | 3.70 | 10.00 | 0.00 | 6.90 |
| MOCA [46] | 22.05 | 28.29 | 5.30 | 14.28 | 19.15 | 28.5 | 3.78 | 13.4 |
| E.T. [17] | 28.77 | 36.47 | 5.04 | 15.01 | 33.78 | 42.48 | 3.17 | 13.12 |
| E.T. + synth. data [17] | 38.42 | 45.44 | 8.57 | 18.6 | 46.59 | 52.82 | 7.32 | 20.87 |
| LWIT [62] | 30.92 | 45.44 | 9.42 | 20.91 | 33.70 | 43.10 | 9.70 | 23.10 |
| HITUT [47] | 21.27 | 29.97 | 13.87 | 20.31 | 25.24 | 34.85 | 12.44 | 23.71 |
| ABP [61] | 44.55 | 51.13 | 15.43 | 24.76 | 42.93 | 50.45 | 12.55 | 25.19 |
| **High-level Goal Instruction Only** | | | | | | | | |
| HITUT G-only [47] | 18.41 | 25.27 | 10.23 | 20.27 | 13.63 | 21.11 | 11.12 | 17.89 |
| LAV [60] | 13.35 | 23.21 | 6.38 | 17.27 | 12.7 | 23.4 | - | - |
| HLSM (Ours) | **29.94** | **41.21** | **20.27** | **30.31** | **29.63** | **38.74** | **18.28** | **31.24** |

**Table 1:** Test results. Test seen/unseen and validation seen/unseen splits. Top section approaches use sequential step-by-step instructions. The bottom section uses only high-level instructions. Best results **using only high-level instructions** and using both types of instructions are highlighted.

**Table 2**

| Method | Validation | | | |
|---|---|---|---|---|
| | Seen | | Unseen | |
| | SR | GC | SR | GC |
| HLSM | 29.6 | 38.8 | 18.3 | **31.2** |
| + gt depth | 29.6 | 40.5 | 20.1 | 33.7 |
| + gt depth, gt seg. | 40.7 | 50.4 | 40.2 | 52.2 |
| + gt seg. | 36.2 | 47.0 | 34.7 | 47.8 |
| w/o language enc. | 0.9 | 8.6 | 0.2 | 7.5 |
| w/o subg. hist. enc. | 29.4 | 38.5 | 16.6 | 29.2 |
| w/o state repr enc. | 30.0 | 40.6 | **18.9** | 30.8 |
| w/o exploration | **32.2** | **42.4** | 18.1 | 31.3 |

**Table 2:** Development results on validation split. Performance of our full approach, with perception oracles, a perception ablation, $\pi^H$ ablations, and $\pi^L$ ablations

**Table 3**

| Task Type | Validation | | | |
|---|---|---|---|---|
| | Seen | | Unseen | |
| | SR | GC | SR | GC |
| Overall | 29.6 | 38.7 | 18.3 | 31.2 |
| Examine | 46.8 | 59.0 | 36.6 | 59.9 |
| Pick & Place | 57.0 | 57.0 | 34.8 | 34.8 |
| Stack & Place | 13.0 | 27.0 | 4.4 | 14.3 |
| Clean & Place | 25.0 | 39.5 | 11.3 | 25.8 |
| Cool & Place | 17.5 | 33.8 | 14.8 | 39.6 |
| Heat & Place | 9.3 | 29.1 | 0.0 | 17.0 |
| Pick 2 & Place | 34.7 | 51.9 | 18.0 | 34.7 |

**Table 3:** Performance breakdown per task type on the validation split.

and unseen scenes, but without a significant difference in performance. Therefore, predicting the sequence of subgoal types and argument classes (i.e., what to do) is at times possible without spatial reasoning, while grounding the subgoal (i.e., where to do it) requires spatial information. Removing random exploration from $\pi^L$ does not significantly affect unseen performance.

Figure 4 illustrates the model behavior, showing both successes and common failures. The main failures in valid unseen scenes are due to (1) perception errors that result in missing or extraneous obstacles or picking up wrong objects; (2) insufficiency of random exploration (e.g., not searching inside cabinets); (3) navigation model errors (e.g., blocking objects from opening); (4) subgoal prediction errors (e.g., picking up wrong objects); and (5) lack of state-aware multi-step planning and backtracking. More qualitative results are available in Appendix **??**.

## 8   Discussion and Limitations

We showed that a persistent spatial semantic representation enables a hierarchical model to achieve state-of-the-art performance on a challenging instruction-following mobile manipulation task. The main performance bottlenecks include long-horizon exploration, perception generalization to unseen environments, and low-level motion planning for continuous collision avoidance. In terms of learning, incorporating reinforcement learning to train $\pi^H$, $\pi^L$, and observation model $F$ jointly could improve robustness. We defined the interface to $\pi^L$ to be faithful to skills available on physical robots, but the exact implementation of $\pi^L$ is not the focus of our work. Physical deployment would require changes to $\pi^L$, and study on robustness to errors in continuous environments, such as localization or motion uncertainty.

# 9 Acknowledgements

This research was supported by ARO W911NF-21-1-0106, a Google Focused Award, and NSF under grant No. 1750499. Animesh Garg is supported in part by CIFAR AI Chair and NSERC Discovery Grant. A significant part of the work was done during the first author's internship at Nvidia. We thank the authors of ALFRED for maintaining the benchmark. We thank Mohit Shridhar and Jesse Thomason for their help answering our questions, and the anonymous reviewers for their helpful comments.

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
