# OpenReview forum: "A Persistent Spatial Semantic Representation for High-level Natural Language Instruction Execution"
_robot-learning.org/CoRL/2021/Conference — CoRL2021 Poster_

### Official Review · Reviewer_s6Fc · 2021-07-23

**Originality:** Good
**Technical Quality:** Very Good
**Clarity Of Presentation:** Good
**Impact:** 3

**Recommendation:**

Weak Accept: I recommend accepting the paper, but will not argue for my recommendation if the majority of other reviewers have a different opinion.

**Summary:**

This work presents an approach to instruction following in which an agent is provided with a high-level natural language instruction of a task and expected to accomplish that task in a previously-unseen environment, discovering and interacting with task-relevant objects as it goes. The addition of a novel "persistent spatial model" (in the form of a semantic voxel grid) helps the agent maintain knowledge of the environment over longer timescales without the need for difficult-to-train recurrent architectures. The voxel grid is built using pretrained segmentation and depth perception models (each of which takes an RGB input to estimate its target output) and the map is fused with language to predict a subgoal---a high-level prediction for the next component of the task---that is then used to guide low-level action selection. Using this approach, the authors achieve state-of-the-art performance, achieving the highest performance to date on the ALFRED dataset for agents not provided with step-by-step instructions at test time.

**Issues:**

Generally, the paper seems mostly complete, I have made a few suggestions in my comments in "Strengths and Weaknesses" above where I think the authors can provide some more context. In summary:

1. The authors should describe in more detail the HiTUT approach that represents their primary point of comparison.
2. The paper should be clearer that, even though step-by-step instructions are not used at test time, they are required to train the system.
3. The rationale behind the "sampling" procedure used during inference should be expanded upon.

**Reviewer Expertise:**

Good: General knowledge of the area

**Strengths And Weaknesses:**

The paper presents a useful and novel approach to improving performance in this domain. Tables 2 and 3 are particularly helpful for both understanding the results and to illustrate where improvements might be made into the future. Though the results are (for the most part) thorough, perhaps the biggest change that I recommend is that it would be particularly helpful if the authors described the HiTUT G-only approach in more detail, since it is the primary approach against which the proposed method is compared. The inclusion of such a description would help to better contextualize the proposed approach. The authors could revisit the motivation for a spatial model, which the HiTUT approach does not have and further reinforce the importance of having a persistent spatial representation.

In the abstract (and perhaps elsewhere), it should be made clearer that though the agent is "avoiding the commonly used step-by-step instructions" at test time, plan subgoals are indeed provided at training time. As there exists other work in which embodied agents are expected to learn their own subgoals, this point should be more clearly communicated. This should not weaken the significance of the work.

[Sec. 4] It would be helpful if the authors could comment on the significance of using "sampling" during inference (perhaps, as opposed to iteratively selecting the next-most-likely subgoal). Is the performance also stochastic, since sampling is randomized, and by how much does performance change for the various tasks if the random seed is changed. I do not think more experiments are necessary here, but some clarification regarding why this process was selected (as opposed to a sequential "maximum likelihood" procedure) and the impact of this decision would help clarity.

Additional Comments and Suggestions:
- Suggestion: the notation "circle with a dot in it" [or \odot in LaTeX] (see footnote 5) is typically reserved for elementwise multiplication. Perhaps using "circle with a \times in it" [or \otimes in LaTeX] is a more appropriate choice, since it often represents the tensor product (similar to the operation defined in footnote 5).
- I believe there is a typo on line 79 (section 3), since T: S x A -> S

**Summary Of Recommendation:**

Overall, the paper is quite thorough. In particular, the detailed studies conducted to understand the importance of each part of the pipeline were a welcome addition, and contributed greatly to the understanding of where this work and other related works in this space may need to improve if the community is to make non-trivial progress. This work represents progress in a challenging application domain that is likely to be used in future work in this area. There are, however, a few places in which the authors should be more clear and help to provide additional context.

---

> ### Author Response · Authors · 2021-08-25
> **Individual Response**
>
> Thank you very much for your helpful and thoughtful comments! We have addressed your suggestions in the paper, and also provide specific comments below:
>
> ### More details on HiTUT
>
> We have added a paragraph in the extended related work section of the appendix that explains the HiTUT method in more detail.
>
> ### Relation to work that automatically extracts subgoals
>
> Thank you for highlighting the ambiguity in how we phrased our claims to not require step-by-step instructions. Our method does require subgoal labels at training, and we added a note in the paper to make this distinction clear. In ALFRED, this segmentation is easy to get from demonstrations, and does not require any annotation effort or complex algorithms.
>
> ### Requiring sequential instructions:
> The step-by-step instructions or demonstration segments from ALFRED are not used at training-time. We use only high-level instructions and full expert demonstration trajectories. The ground truth subgoals are automatically inferred from the demonstration sequences by taking, at timestep t, the next interaction action executed by the expert as the current subgoal at time t.
>
> ### State of the art results:
> See the common response. The state-of-the-art performance is absolute, even compared to approaches that do use the step-by-step instructions.
>
> ### Significance of using sampling:
> There are two reasons to prefer sampling over a next-most-likely approach (i.e., argmax).
>
> First, there are two types of subgoal execution failures: systematic and random. An example of a systematic failure is the selection of an incorrect subgoal. For example ToggleObjectOn(FloorLamp) would fail if a FloorLamp does not exist in the environment. An example of a random failure is the low-level controller sampling an interaction pose for which the interaction fails (e.g. Figure 4, row 1, timestep 272).
> A next-best approach would alleviate a systematic failure, but a sampling approach alleviates both: the systematic failures by trying different subgoals, and random failures by potentially sampling the same subgoal multiple times.
>
> Second, between subsequent repeated subgoal attempts, the agent moves through the environment and updates the state representation. This can alter the probability ordering of subgoals and complicate using a next-best approach.
>
> ### Is the performance stochastic:
> Yes, the performance is stochastic. Our subjective experience during development is that the valid_unseen set (n=821) is large enough that subsequent attempts with the same model vary within less than ~+-1%. We posit that performance on test_unseen (n=1529) would vary less.

---

### Official Review · Reviewer_h53W · 2021-07-24

**Originality:** Very Good
**Technical Quality:** Very Good
**Clarity Of Presentation:** Excellent
**Impact:** 4

**Recommendation:**

Strong Accept: I recommend accepting the paper and will argue for my recommendation even if other reviewers hold a different opinion.

**Summary:**

This paper presents a spatial semantic representation method that enables an agent to reason about and plan behaviors to complete long-term tasks. The developed method called hierarchical language-conditioned spatial model (HLSM) connects vision, language, and control that are three important topics toward developing general AI. The input include high-level language instructions and raw images, and the output are low-level mobile manipulation actions. The developed approach was evaluated using the recent ALFRED benchmark. HLSM was compared to a recently published approach called HiTUT, and methods that process detailed language-based instructions. Results show improvements in success rate.

**Issues:**

See above.

**Reviewer Expertise:**

Good: General knowledge of the area

**Strengths And Weaknesses:**

The reviewer didn't observe any major issues from the paper. Technical details were well presented, and it's quite easy to follow the paper. The baseline methods were carefully selected, and the results are convincing. There are the following suggestions that can potentially further improve the work.

The evaluation and implementation highly depend on ALFRED. Although it's a well developed benchmark, all benchmarks have limitations and ALFRED is no exception, which was not discussed. It's necessary to discuss how the developed methods can be applicable to real-world scenarios. There are assumptions such as reliable action executions that won't be valid for real robots.

It's unclear what makes the learned spatial representation "persistent". The term is highlighted in the title, and throughout the paper, but was not explained.

All components were learned using supervised learning methods. The datasets were separately collected, and the neural networks were learned separately. This type of "piece by piece" learning requires a lot of fine-tuning work. Also, it's unclear which part is the bottleneck of the current system. Some more discussions will be appreciated.

**Summary Of Recommendation:**

The reviewer didn't observe any major issues from the paper. It's a nicely written paper of solid research.

---

> ### Author Response · Authors · 2021-08-25
> **Individual Response**
>
> Thank you for your review and for identifying areas of clarification and improvement!
>
> We have addressed your concerns of generalization to real robots in the common response, and also added a section in the Appendix. We have expanded the discussion with limitations and mentioned the bottlenecks. We provide additional clarifications below:
>
> ## Clarifications
>
> ### Piece-by-piece learning requires fine-tuning:
> While we show the advantages of the modularity of our approach, ideally we would optimize these modules jointly to avoid the need for fine-tuning. This is a challenging learning problem at this scale, both in terms of formulating it as an optimization problem, and solving it with a practical algorithm. We leave it for future work. That said, the modular approach allows tuning the individual modules in isolation, which can be more intuitive than hyperparameter tuning on large neural networks or complete systems.
>
> ### Meaning of “Persistent”:
> Semantic and occupancy information about a location that is projected and added to the semantic voxel map remain there until a subsequent observation of the same location, even if many timesteps later. This allows the agent to operate continuously in the environment for hundreds of timesteps, and “remember” objects seen earlier in the execution.

---

### Official Review · Reviewer_Xmxc · 2021-07-26

**Originality:** Good
**Technical Quality:** Very Good
**Clarity Of Presentation:** Good
**Impact:** 3

**Recommendation:**

Weak Reject: I recommend rejecting the paper, but will not argue for my recommendation if the majority of other reviewers have a different opinion.

**Summary:**

The paper proposes augmenting hierarchical mobile manipulation policies with a persistent semantically-labeled voxel map (spatial semantic representations) learned from natural language instructions. It seeks to show that conditioning a hierarchical policy on these persistent representations allows agents to succeed at high-level mobile manipulation tasks.

Proposed contributions:
* "First" demonstration of a modular representation learning approach for the problem of mapping high-level NL task descriptions to actions in a 3D environment
* Method f or utilizing spatial semantic representations within a hierarchical model for solving MM tasks
* SOTA performance on ALFRED with only high-level instructions (rather than step-by-step instructions)

**Issues:**

* Spending more time on building intuition, teaching the reader about the problem, and telling us what insights about the problem we can take from the method
* Better-explaining why the method works, and why it fails
* Adding (even a small) evaluation on some challenge other than ALFRED

Detailed Comments
---
Figure 2: This figure is very dense and takes a lot of work to parse. I suggest splitting it into two or more figures for clarity, each representing a single idea.

Line 239: "...even though this evaluation is not fair to our method" -- Fairness is very much in the eye of the beholder and really depends on what the reader is looking for in a method. I suggest instead noting that evaluating on step-by-step instructions is *different* and won't necessarily demonstrate all of your method's potential. Similarly for Line 251.

Line 252: Whether or not the authors of ABP have published the details of their method is irrelevant to your performance comparison. Similarly with LAV. Notes like this (and greying them in the table) come off as unprofessional and a bit petty.

Table 1: What is the purpose of bolding the line for HLSM? Bolding is sometimes used to call out the highest numbers in a column in ML/CV/NLP papers, but that doesn't seem to be the case here and there is no explanation in the caption. Please consider removing this. Similarly with Tables 2 and 3.



**Reviewer Expertise:**

Good: General knowledge of the area

**Strengths And Weaknesses:**

Strengths
---
* Clear, unambiguous, and generally high-quality presentation
* Results and figures included are helpful for understanding the method and thorough
* Method improves on SOTA by eliminating the need for step-by-step instructions to achieve current success rates

Weaknesses
---
* The proposed architecture is exceedingly complex, and seems highly-customized to a specific benchmark (ALFRED). I would like to see an application to at least one other benchmark to be assured that this system is generally-applicable. The complexity also make it unlikely future authors will be able to successfully build on the method.
* Results seem to support the assertion that HLSM allows this system to meet the performance of other methods which are provided with step-by-step instructions rather than high-level instructions (a triumph in and of itself!) but not necessarily improving SOTA on ALFRED
* The paper spends so much time on very detailed descriptions of the method that it leaves little space for building intuition behind the method, analysis, or discussion --- i.e. it is a description of a method, plus results, plus a declaration of beating SOTA. The reader will not learn much about the problem or future possibilities by reading this paper.

**Summary Of Recommendation:**

My recommendation for a weak reject is motivated by a couple of my comments above, specifically:

(a) I'm concerned this work as-presented may be overfitted to the ALFRED benchmark, leaving me with few reasons to believe it might work if applied to real robots and

(b) while a thorough and convincing description of a method and results, I don't think the reader will learn much about the core research problem, future possibilities for new research, or build an understanding as to why this method succeeds or fails.

---

> ### Author Response · Authors · 2021-08-25
> **Individual Response**
>
> Thank you for your valuable comments and suggestions!
>
> # Revisions:
> - We updated the text to avoid phrasing the sequential instruction evaluation as being “unfair” to our method. Thank you for pointing out how it appears to the reader.
>
> - We amended the paper to remove the greying out of ABP and LAV results. Thank you for pointing out that it comes across negatively. The greying was intended only to communicate why we discuss only HiTUT (#3 on leaderboard) instead of ABP (#2 in the leaderboard) in our discussion, but there are better ways to achieve that.
>
> - The bolding in results tables refers to the highest performance in the evaluation setting of our interest, mapping high-level instructions to actions. We clarified this in the caption. For readers who are primarily interested in the ALFRED benchmark evaluation setting, we also highlighted the best results there using underlines. We hope this is more informative.
>
> - We have split Figure 2 into two separate figures and replaced the flowchart with the corresponding algorithm.
>
> - We have added an extended related work section in the Appendix that explains the problem domain and research in the area in more detail, and motivates our approach in that context. We also revised the text to add motivating sentences, as much as page limit allows without cutting crucial technical details.
>
> # Clarifications:
>
> ### Method Complexity
> In terms of the set of abstractions used and individual modules with underlying implementations, the complexity of our method is similar to other methods in robotics (e.g. [3]). When compared to methods that also follow instructions from raw observations (e.g. [1]) (albeit not mobile manipulation instructions in household environments - such capabilities don’t exist yet), our method is similar or even simpler, in a sense of using less individually engineered or learned components. In our description, we emphasize technical depth to allow for maximal technical understanding and insights within the page limit. We understand this may cause our method to seem more complex when compared to papers that provide only high-level description of their technical contribution. If there are specific components that seem overly complex, we are happy to discuss each such specific issue.
>
> One could argue that our system is more complex than e.g. end-to-end transformer architectures (such as [2]). Such architectures are also exceedingly complex and followed years of research before being included in popular libraries, but more importantly, end-to-end approaches have shown limited evidence of capability to follow high-level natural language instructions in embodied mobile manipulation environments in a way that has an avenue for future real-world deployment. This does not mean this won’t happen, but likely does require further work, which our work aims to facilitate.
>
> ### Ability of future work to build on this method:
> We will open-source the codebase with clearly delineated classes and abstractions for the high-level and low-level controllers, state representation, action, observation model, and all procedures within the low-level controller. The modularity of our approach makes it easier for future authors to build on the method in a way that end-to-end neural network methods don't.
>
> Follow-up work on ALFRED (or similar future benchmarks as they are released) interested in specific research problems such as exploration, perception, high-level planning, navigation, language grounding, etc, can use our codebase and the reusable modules that it provides, and focus efforts on problems of their interest, without having to solve the entire instruction-following problem, which takes a significant amount of effort. We believe that our codebase will effectively reduce the barrier of entry for research on long-horizon planning and following high-level mobile manipulation instructions.
>
> ### State-of-the-art results
> See the common response.
>
> ### Concerns on lack of other benchmarks
> See the common response.
>
> ### Generalization to real robots:
> See the common response.
>
> ### References:
> [1] Patki, Siddharth, et al. "Language-guided semantic mapping and mobile manipulation in partially observable environments." CoRL (2020).
> [2] Pashevich, Alexander, Cordelia Schmid, and Chen Sun. "Episodic Transformer for Vision-and-Language Navigation." arXiv preprint arXiv:2105.06453 (2021).
> [3] Garrett, Caelan Reed, et al. "Online replanning in belief space for partially observable task and motion problems." ICRA (2020)

---

### Official Review · Reviewer_xkZt · 2021-08-02

**Originality:** Good
**Technical Quality:** Good
**Clarity Of Presentation:** Fair
**Impact:** 3

**Recommendation:**

Weak Accept: I recommend accepting the paper, but will not argue for my recommendation if the majority of other reviewers have a different opinion.

**Summary:**

The paper considers the problem of executing high-level commands provided via natural language, which typically requires reasoning over extended time horizons. Modeled in similar spirit to a partially observable MDP, there are two aspects to the proposed framework that are fundamental to addressing the problem. One is a 3D spatial-semantic model of the environment that is maintained over time based upon visual (monocular) observations, which the paper argues as providing a persistent representation of the environment. The second is a hierarchical representation of the action space, whereby language is mapped to (a distribution over) abstract subgoals based on the state representation along with the subgoal history. A low-level policy then maps a particular subgoal to an action based on the current state. The method is evaluated on the ALFRED benchmark and shown to outperform a recent method (HiTUT) that also reasons over high-level instructions. An ablation study demonstrates the contributions of the different architecture components on seen and unseen environments.

**Issues:**

* Lines 64–65: The statement that "laborious to scale to new environments and complex language" is questionable, particularly given the success of symbol grounding-based approaches to language understanding in robotics. It is the symbolic representation of the state and actions that enable generalization to new environments and the exploitation of the hierarchy of language (e.g., [1 in the paper]) that supports the interpretation of complex language.
* How does the spatial-semantic representation of the environment proposed here compare to representations used by others in the context of language understanding [37(paper), 2]
* How are the state dynamics modeled? Are they assumed to be known or are they learned?
* Lines 133–134: How exactly is the perception step modeled to support training from a limited amount of data?
* Lines 141–142: What are the segmentation distributions, exactly?
* The approach relies upon a dead-reckoned estimate of the robot's pose, which will quickly drift in practice. How does this affect the projection of the environment representation? Regardless of projection, how does this influence the environment model, which, as far as I can tell, does not account for the pose uncertainty.
* Can the authors elaborate on the reduced performance on unseen environments when including the state encoding? Was the model with or without state encodings used for the test set (i.e., did this function as a validation set in terms of choosing the architecture)?


REFERENCES

[1] Siddharth Patki, Ethan Fahnestock, Thomas M. Howard, and Matthew R. Walter. Language-guided Semantic Mapping and Mobile Manipulation in Partially Observable Environments, In Proceedings of the Conference on Robot Learning (CoRL), 2019.

**Reviewer Expertise:**

Excellent: Expert knowledge on the topic of the paper

**Strengths And Weaknesses:**

STRENGTHS

+ The problem of executing natural language instructions that provide only a high-level description of the task is challenging and, as the authors note, has received less attention recently when compared to methods that reason over the more detailed, lower-level (step-by-step) instructions.
+ The method is evaluated on a standard, albeit simulated, benchmark and. shown to outperform a recent high-level baseline.

WEAKNESSES

- While the body of work that is cited is reasonable, the discussion of how this work is situated with regards to the large body of work in language understanding for robotics is inadequate. At only two paragraphs, the related work section provides only a cursory description of a small fraction of work in language understanding, in many cases, without a discussion of how the proposed approach differs. This makes it difficult to assess the significance of the contributions. This is not the first paper to consider spatial-semantic environment models in the context of language understanding nor is it the first to consider mapping language to high-level abstractions (symbols) of the environment and the robot's actions. That said, I think that the work is interesting and potentially novel---the paper just needs to do a better job of making this evident.
- The paper initially presents a formulation of the problem as a stochastic process, but it is not always clear how the proposed framework is consistent with formulation (e.g., it isn't clear whether or how the prediction and update step are principled, or what the prediction step is, for that matter).
- Related, the technical details of the algorithm can be difficult to parse at times, in part because various details are omitted. As presented, some of aspects of the algorithm come across as ad hoc.
- It isn't clear how well the method will scale to larger environments
- It is unclear whether the approach can be expected to generalize to real-world settings (e.g., including the ability to handle uncertainty in perception and motion).


MINOR

* Line 66: "... language directly actions ..." --> "... language directly to actions ..."
* Figure 2: Where is s_t? Also, it would help to use the same notation as in the text.
* Line 156: Erroneous comma before footnote.


**Summary Of Recommendation:**

The paper considers an important problem, namely the ability to interpret and carry out high-level instructions. The results indicate that the method outperforms a recent baseline, however the qualitative comparison to related work in language understanding for robotics is lacking, which makes it difficult to judge the contributions relative to existing work. With an evaluation that is limited to simulation, it is also difficult to predict how the approach would generalize to the real world, where the uncertainty is presumably far greater than in the simulator.


UPDATE BASED ON AUTHOR RESPONSE

I appreciate the authors' detailed response, which clarified several of the questions that I had. I remain concerned that the related work discussion doesn't adequately identify the paper's contributions relative to existing work. Several citations have been added to the related work section, but they are only given a cursory description in the primary paper. While the appendix has been updated to include more details, the paper's contributions need to be clearly presented as part of the main text.

---

> ### Author Response · Authors · 2021-08-25
> **Answers and Clarifications**
>
> ## Answers and Clarifications.
>
> >How does the spatial-semantic representation of the environment proposed here compare to representations used by others in the context of language understanding [37(paper), 2]
>
> We expanded the related work section, and added a paragraph on “Semantic maps for language grounding in robotics” to the extended related work section in the appendix (that is not constrained by the page limit) that elaborates on the differences.
>
> > How are the state dynamics modeled? Are they assumed to be known or are they learned?
>
> The GoTo procedure in the low-level controller is based on a value-iteration network that utilizes a deterministic grid-navigation dynamics model on the internal representation, which is a crude (but simple) approximation of the dynamics of the RotateLeft, RotateRight and MoveAhead navigation commands. Other than that, the dynamics of the environment are unknown to the agent, and are not explicitly modeled.
>
> > Lines 141–142: What are the segmentation distributions, exactly?
>
> In Line 141-142, we refer to the per-pixel segmentation distributions defined in Line 130, for all points in each voxel. There is a 1:1 mapping between pixels and points.
>
> > Effects of pose uncertainty on the approach.
>
> Our approach assumes reliable pose estimates. The dead-reckoning is merely a means to obtain pose estimates in ALFRED, and we do not suggest it as viable for a real robot, where better localization sources exist. Intuitively, voxels further away are affected by pose estimate errors more, but are likely more tolerable due to being used mainly to decide navigation goals. Voxels close to the agent require more precision as they are used for object instance mask generation, but would be less affected by pose estimate errors. Our voxel map uses a relatively coarse 25cm resolution.
>
> > … reduced performance on unseen environments when including the state encoding?
>
> During training, the high-level controller was trained on data with perfect segmentation, and thus with state encodings that are perfect representations of what object classes have been observed by the agent. In unseen environments, the semantic segmentation is erroneous due to the generalization gap, resulting in erroneous state encodings. We hypothesize that this difference between training and test-time is the main cause for the performance drop in unseen environments. The generalization gap is mostly visual, and the other inputs (language and subgoal history) are affected less.
>
> > Which model was used for the test results?
>
> The model with the state-encodings was used for the test results (i.e. this did not function as a validation set). We performed only a single run on the test-set, and thus have not tested a variant without the state-encodings to avoid overfitting, but the validation results hint to a potential to achieve even better test results.
>
> > Ability to handle uncertainty in perception and motion.
>
> While we do assume accurate pose estimates, ALFRED is actually a very challenging test of handling perception errors. Given only 108 artificial training scenes, each with a fixed set of objects, lighting and layout, it is challenging to learn a perception model that generalizes to the visuals of previously unseen scenes that contain many completely new textures, layouts, and objects.
> See videos with
> [learned vision](https://www.youtube.com/watch?v=J26kuDAbDLI)
> and
> [ground-truth vision](https://www.youtube.com/watch?v=fSTKYBWkVV0)
> in the same environment from the valid_unseen set.
> The ability of our approach to achieve state-of-the-art on unseen environments demonstrates better perceptual generalization compared to prior work.
>
> > On “ad-hoc” nature of some aspects of the algorithm.
>
> We understand how the low-level controller (LLC) design specifically may come across as ad-hoc. The low-level controller stack is not our focus, but a necessary ingredient to show experimental results with the complete system. If there are other aspects that come across as ad-hoc, please let us know and we will do our best to clarify.
>
> > Relations to stochastic processes, update and prediction steps.
>
> Executing our policy in the ALFRED environment amounts to a stochastic process, however we did not use the concepts of update and prediction steps, because our policy execution flow does not precisely adhere to such a formulation. We hope the addition of Algorithm 1 clarifies the computation.
>
> > Scaling to larger environments
>
> We assume “larger” refers to the physical scale of the world. Many practical robotic domains (e.g. household environments as studied here) are relatively small, and an advance in such environments is significant in its own right. The main bottleneck towards scaling to larger environments is the memory constraint of the semantic memory. This is an important direction for future work, for example by using multi-scale representations such as Octrees.
>
> > Generalization to real robots:
>
> See the common response.

---

> ### Author Response · Authors · 2021-08-25
> **Individual Response**
>
> Thank you for your feedback and comments, and for appreciating the challenge and importance of the problem we study!
>
> ## Revisions:
> - We expanded the related work section to give credit to relevant approaches on semantic mapping and language grounding for robotics. We also added an extended related work section in the Appendix unconstrained by the page limit, and an FAQ section that, among others, addresses your questions.
>
> - Regarding the stochastic process formulation: we have replaced the Execution Flow in Figure 2 with Algorithm 1, which explains the execution flow of the system more precisely and connects all the pieces better. We will also release an extensible and modular code to let others build upon our work, including our low-level controller for the ALFRED environment. Please let us know if there is further need for clarification.
>
> - Lines 64-65: we rephrase the statement to better convey our intended meaning. Symbolic representations provide stability and robustness, and are an excellent choice for many robotics applications, but the ontology that they rely on can be hard to scale to unstructured environments, such as households.
>
> - Lines 133-134: The depth and segmentation models use relatively lightweight U-Net models that are much smaller than recent SOTA segmentation and depth models such as Detectron2 [53 in revised paper]. We have rephrased this statement.

---

### Meta-Review · Area_Chair_zma6 · 2021-08-13

**Recommendation:** Accept (Poster)
**Confidence:** 5

**Metareview:**

This approach to an important problem appears to have achieved a substantial advance over SOTA, though there is some concern about overfitting to the ALFRED benchmark, which it would be helpful for the authors to discuss.

 — post-rebuttal, I believe many of the questions surrounding the contribution and method have been resolved, and the reviewers are largely positive about the paper. One reviewer is still a weak reject, but did not respond to the authors’ rebuttal.

---

> ### Author Response · Authors · 2021-08-25
> **Common Response**
>
> # Common response:
>
> We thank the reviewers and AC for the detailed comments. We are excited that all reviewers appreciate the presentation quality, find the task important and challenging, our approach interesting, and the results impressive. Below, we address common core questions raised. We also provide individual responses in separate posts to make sure all questions are answered.
>
> We are looking forward to further answering any questions raised and discussing the work in the remainder of the response period.
>
>
> # Common Clarifications:
> ## Contribution and scope
>
> Our focus and main contributions are the spatial-semantic representation and empirical results that show how a hierarchical policy can utilize such a representation to achieve state-of-the-art results on a challenging language-guided mobile manipulation benchmark.
>
> ### Requests for additional benchmarks
> We generally agree with reviewer Xmxc about the benefit of an additional benchmark. To our knowledge, there are no other benchmarks displaying this type of high-level language in an interactive 3D environment with raw first-person observations. This is likely because such tight integration between language, robotics, and perception is relatively nascent. All prior work on the ALFRED benchmark does not evaluate on other benchmarks either [12, 16, 17, 18, 45, 49, 61 in revised paper]. As the field progresses, we are certain more benchmarks will be introduced. There is a chicken-and-egg problem here: for more benchmarks to come along, this topic must show dynamic research that is accepted by the community, which depends on the top venues displaying such research. We hope our work contributes to this progression.
>
> ### State-of-the-art results
> Our approach is currently #1 on the official ALFRED leaderboard, which is judged according to the success rate on the test_unseen split.This is across all methods, regardless if they use sequential instructions or not. Our approach improves SOTA on ALFRED in absolute terms (not only among methods that use only high-level instructions). Here are the current top-5 entries on the ALFRED leaderboard (we are not including a link to avoid breaking anonymity):
>
> | Rank | Method | SR (test_unseen) | GC (test_unseen) |
> |------|--------|------------------|------------------|
> | 1    | HLSM   | 0.1629           | 0.2724           |
> | 2    | ABP (Kim et al.) | 0.1543 | 0.2476           |
> | 3    | HiTUT (Zhang et al.) | 0.1387 | 0.2031       |
> | 4    | LWIT (Nguyen et al.) | 0.0942 | 0.2091       |
> | 5    | Episodic Transformer (Pashevich et al.) | 0.0857 | 0.1856 |
>
> We have updated the results table to more accurately communicate the state of the leaderboard.
>
> ### Clarifications and Questions
> We added a section of frequently asked questions in the Appendix, to answer common questions, including some of these raised by the reviewers. This is an addition to the individual responses below.

---

> ### Author Response · Authors · 2021-08-25
> **Generalization to real robots**
>
> ## Generalization to real robots
>
> The observation model, high-level controller, state representation, and the interface to the low-level controller together constitute our contribution and are intended to generalize to physical robots. Both ALFRED and our approach are designed to interface with robotic systems (e.g., ALFRED uses masks for object selection). Deployment on a physical robot would require an implementation of the low-level controller designed for continuous motion in cluttered environments, and an implementation of the ALFRED interface to enable execution of manipulation actions such as “Pickup” and “ToggleOn” conditioned on object instance masks. Such physical robot capabilities/interfaces are subject of significant ongoing research. This provides an avenue towards real-robot deployment of our approach.
>
> Real-robot operation is the long-term motivation of this work and has been carefully considered in the  approach (and also in the design of ALFRED by its authors). We do not claim to be able to execute high-level natural language mobile manipulation instructions from raw vision on real robots in unseen environments just yet though. To date, such capabilities haven’t been demonstrated even in simulated environments. ALFRED is presently the simplest “hard problem” that enables meaningful progress. Even though our method achieves better results than all existing work, it can still only solve 16% of problems in unseen environments. So, while we can do things we couldn’t before (i.e., act in ALFRED with only high-level instructions), there is still much work to be done.
>
> Following high-level mobile manipulation instructions is a challenging problem where impactful progress can be made in simulated environments, deferring the high cost of real-robot integration to a future time when the methods show better success rates. A similar research path towards real-robot instruction-following has previously seen success on indoor mobile robots [5 -> 6 -> 7] and quadcopters [8 -> 9 -> 10] with representation learning, and a mobile manipulator forklift [11 -> 12 -> 13] using symbolic representations.
>
> ### Assumptions relevant to real-robot deployment:*
> - Low-level controller: Our approach assumes a low-level controller (LLC) stack capable of executing the interaction subgoals, by navigating to pose goals and generating segmentation masks as input to a manipulation policy. For this paper, we implemented an LLC specifically for AI2Thor. Our design is faithful to resources available on physical robots, such as 2D navigation policies and localization. At the API-level (subgoals as input, actions and object masks as output), existing work on navigation (e.g. [1, 2]) and manipulation (e.g. [3, 4]]) points to possibilities of implementing the LLC on a real robot.
>
> - Localization: Our approach assumes access to reliable localization. On physical robots, this could be achieved using motion capture, visual odometry, environment instrumentation, LIDAR mapping, or a mix. As localization is a well-studied problem in robotics, we see this as a reasonable assumption for the time being. We refrain from making claims about robustly handling diverse localization noise, because it’s extremely challenging to simulate in an environment like ALFRED without introducing biases.
>
> - Reliable action execution: ALFRED does not guarantee reliable action execution (e.g. placing and opening objects, and navigation is subject to collision constraints). The real world is still far more noisy and uncertain. An improved low-level controller would be required to handle this uncertainty, which is orthogonal to our work.

---

> ### Author Response · Authors · 2021-08-25
> **Common Response References**
>
> [1] Wijmans, Erik, et al. "DD-PPO: Learning near-perfect pointgoal navigators from 2.5 billion frames." ICLR (2020).
> [2] Chiang, Hao-Tien Lewis, et al. "Learning navigation behaviors end-to-end with autorl." IEEE Robotics and Automation Letters 4.2 (2019)
> [3] Xie, Christopher, et al. "Unseen object instance segmentation for robotic environments." IEEE Transactions on Robotics (2021).
> [4] Lenz, Ian, Honglak Lee, and Ashutosh Saxena. "Deep learning for detecting robotic grasps." IJRR (2015)
> [5] Anderson, Peter, et al. "Vision-and-language navigation: Interpreting visually-grounded navigation instructions in real environments." CVPR (2018).
> [6] Tan, Hao, Licheng Yu, and Mohit Bansal. "Learning to navigate unseen environments: Back translation with environmental dropout." NAACL-HLT (2019).
> [7] Anderson, Peter, et al. "Sim-to-real transfer for vision-and-language navigation." CoRL (2020).
> [8] Misra, Dipendra, et al. "Mapping instructions to actions in 3d environments with visual goal prediction." EMNLP (2018).
> [9] Blukis, Valts, et al. "Mapping navigation instructions to continuous control actions with position-visitation prediction." CoRL (2018).
> [10] Blukis, Valts, et al. "Learning to Map Natural Language Instructions to Physical Quadcopter Control Using Simulated Flight." CoRL (2019).
> [11] Kollar, Thomas, et al. "Toward understanding natural language directions." HRI. (2010).
> [12] Tellex, Stefanie, et al. "Understanding natural language commands for robotic navigation and mobile manipulation." AAAI. (2011).
> [13] Walter, Matthew R., et al. "A situationally aware voice‐commandable robotic forklift working alongside people in unstructured outdoor environments." Field Robotics (2015).

---

### Decision · Program_Chairs · 2021-09-13

**Decision:**

Accept (Poster)

**Comment:**

This approach to an important problem appears to have achieved a substantial advance over SOTA, though there is some concern about overfitting to the ALFRED benchmark, which it would be helpful for the authors to discuss.

 — post-rebuttal, I believe many of the questions surrounding the contribution and method have been resolved, and the reviewers are largely positive about the paper. One reviewer is still a weak reject, but did not respond to the authors’ rebuttal.